# Creating Actionable and Insightful Knowledge Applying Graph-Centrality Metrics to Measure Project Collaborative Performance

**Marco Nunes** [1,*], **Jelena Bagnjuk** [2], **António Abreu** [3,4], **Edgar Cardoso** [5], **Joana Smith** [6] and **Célia Saraiva** [7]

1 Project Management Department at Tetra Pak, Wilhelm-Bergner-Straße 9c, 21509 Glinde, Germany
2 Project Management Department, University Medical Center Eppendorf, Martinistraße 52, 20251 Hamburg, Germany; j.bagnjuk@uke.de
3 Department of Mechanical Engineering, Polytechnic Institute of Lisbon, 1959-007 Lisbon, Portugal; ajfa@dem.isel.ipl.pt
4 CTS Uninova, 2829-516 Caparica, Portugal
5 Senior Data Analyst at Deutsche Bank, AG 1 Great Winchester Street, London EC2N 2DB, UK; edgar.c.nunes@gmail.com
6 Supply Chain Management Department at Borgwarner, 3000 University Drive, Auburn Hills, MI 48326, USA; joanasmth1@gmail.com
7 Department of Informatic Engineering, UTAD-IST, Quinta de Prados, 5000-801 Vila Real, Portugal; celia.saraiva@gmail.com
* Correspondence: marco.nunes@tetrapak.com or nunesmr@gmail.com

**Abstract:** Tools and techniques supported by math and statistics are often used by organizations to measure performance. These usually measure an employees' traits and states performance. However, the third type of data usually neglected by organizations, known as relational data, can provide unique and actionable insights regarding the root causes of individual and collective performance. Relational data are best captured through the application of graph-based theory due to its ability to be easily understood and quantitatively measured, while mirroring how employees interact between them as they perform work-related tasks or activities. In this work, we propose a set of graph-based centrality metrics to measure relational data in projects by analyzing the five most voted relational dimensions ((1) communication, (2) internal and external collaboration, (3) know-how exchange and informal power, (4) team-set variability, and (5) teamwork performance), in a survey conducted to 700 international project stakeholders in eight business sectors. The aim of this research is to tackle two issues in projects: First, to understand in a quantitative way how the project's relational data may correlate with project outputs and outcomes, and second, to create unique and actionable knowledge to help mitigate the increasing project failure rates. A case study illustrates the step-by-step application of the developed graph-based metrics as well as its benefits and limitations.

**Keywords:** project management; graph-centrality metrics; project outcome; project lifecycle; individual performance; collective performance; correlation

## 1. Introduction

In almost every organization, parallel to the formal organizational chart, a different type of network naturally emerges and evolves, characterized by the mix of informal and formal relationships between employees [1–3]. According to several studies, such a mix is usually responsible for how work gets done in organizations [2–6].

Informal relationships, also known as informal organizational networks, are often hidden behind the typical organizational formal chart and can only be uncovered by the application of graph-based metrics [1–3,5]. Such a mix of formal and informal networks is also known as relational data [1,4].

Research shows that it is practically impossible to clearly distinguish the formal from informal relationships that exist within an organization's social network, however, because

informal relationships usually extrapolate the formal ones, the mix of formal and informal relationships is usually called informal relationships or informal networks [7,8].

Relational data can be collected through the application of strategic surveys, strategic observations, and consulting data logs [1,2]. It is measured by applying graph-based metrics, such as in/out or average degree, closeness, betweenness, just to name a few [1,5,9–11]. The results can then be used to evaluate individual and collective performance, among other important organizational factors [12,13].

According to several studies, graph-based network centrality metrics are those that provide the most valuable insights when analyzing organizational informal networks [1,2,9–12]. Such centrality metrics measure in a quantitative way how important a given entity or a group of entities (people, groups, or organizations) within a social organizational network are. This is measured by analyzing collaborative dimensions, such as information sharing, communication, know-how exchange, and problem-solving, among many others [10–13]. For example, in an organizational context, employee centrality is understood as the advantage of being in a position within the organization that is not far from everybody, and as a consequence of it, an employee tends to ear things first and faster than others, allowing that employee to get a deeper insight into what is going on around the organization [1,2]. Furthermore, research shows that the application of graph-based centrality metrics enables organizations to identify informal key players that highly impact an organization's outputs outcomes [7,13]. Such key informal players include: Central hubs—people that are highly connected within a social network, information brokers—people that connect different pockets of an organization, boundary brokers—people that bridge different organizations or departments, peripheral—people that are at the outskirts of a social network either because they are not well integrated or because they may have some kind of expertise that requires some kind of isolation, energizers/de-energizers—people that positively or negatively influence a team or a group regarding the execution of organization's tasks and activities for example [13,14].

Because it is hard to find something that is not somehow connected, it is critical for an organization to uncover such hidden relationships. The reason for this is that it enables organizations to take the appropriate measures in order to properly manage such hidden relationships and thus potentially boost their overall organizational performance by uncovering pain points and leveraging hidden critical connections and capabilities [1,7,13–16].

Centrality graph-based metrics can be divided into two major groups [9]. They are: (1) Node (individual), and net (2) work (collective). Node centrality metrics include, but not only, in/out/total, and average degree, closeness, betweenness, and clustering degree. Network centrality metrics include, but not only, average in-degree/out-degree/total-degree, density, distance, and average distance. However, many existing graph-based centrality metrics do not capture the evolution of relational data in project environments in isolation. For example, the in and out-degree metrics alone are not enough to explain correlations between complex relationships and project outputs and outcomes. Due to this reason, the development of new, or the adaptation of existing graph-based centrality metrics supported by other measuring tools and techniques, such as basic statistics, work more efficiently in identifying dynamic relationships between project outcomes and outputs and informal networks.

In this work, we propose a set of graph-based centrality metrics to quantitatively measure project relational data by analyzing five key relational dimensions (KRD) that will help organizations to measure individual and collective relational performance and understand the potential correlation between informal networks and project outputs and outcome. They are: (1) Communication, (2) internal and external collaboration, (3) know-how exchange and informal power, (4) team-set variability, and (5) teamwork performance). These five KRD result from a survey conducted with 700 international project stakeholders between 2018 and 2021, where participants were asked to name some of the most important project collaborative dimensions. The necessary data to build each one of the KRD can be collected through project surveys, project meetings, and consulting project logs.

The aim of this research is to simultaneously tackle two issues in project management. First, to understand in a quantitatively way how projects' informal relationships may be correlated with project outputs and outcomes, and second—as a direct consequence of the analysis –, provide a complementary contribution to help understand the still increasing project failure or challenge rate as shown by some of the most internationally renowned project institutes, such as the PMI (project management institute) and the Standish Group [17–19].

This work is divided into six sections. In section one, a brief introduction is presented to highlight the importance of quantitatively measuring relational data in organizations and the main objectives of this research. Section 2 presents an extensive literature review on state of the art regarding the major topics addressed in this work. Section 3 introduces the fundamentals regarding the research and development of proposed key metrics. Section 4 introduces the proposed key metrics, and a case study explains in a step-by-step approach the calculation process of proposed key metrics. Section 5 is a discussion regarding the results obtained in the previous section and the respective academic and managerial implications. Finally, Section 6 presents the major conclusions and suggests further steps toward the improvement of the present research.

## 2. Literature Review

According to several studies, projects keep failing at an impressive pace, though the number of organizations, institutes, and bodies of knowledge that provide project guidance has exponentially increased over the recent years [17–20]. Two of the most renowned organizations that monitor the evolution of how projects evolve throughout the years (the PMI and the Standish Group), in their public reports, show that the results are far from positive. In the PMI's Pulse of the Profession report that covers a period from 2011 to 2019, the number of projects that experienced any type of scope creep rose from 41% in 2011 up to 53% in 2019. Furthermore, the number of projects with failed project's budgets has remained relatively constant at 35% between 2011 and 2019 [21,22]. Moreover, according to the report of the Standish group for the period between 2017 and 2020, the number of projects that failed reached a total of 19%, while the number of projects that experienced any type of failure or challenge reached up to 50%.

Although there may be many reasons behind the number of both organizations that lead projects to fail or become challenged, research points out three major areas that need further research [23–25]. They are: (1) Processes—the different existing project management processes according to institution or organization and how they are understood by different people in managing projects, (2) principles—the way project management rules and best practices are understood by different people, organizations, cultures etc., and (3) people—how the way project stakeholders work together to execute project tasks and activities impacts project outcomes and outputs.

In this work, the people aspect is addressed. Several studies show that more than individual skills and expertise, the ability to work efficiently together within a group or organization is twice a predictor of success [1,6,13,23,26]. For this reason, it is critical to address the people aspect to quantitatively understand how the way that people work together in projects may or not be correlated with project outcomes. The way people work together (also known as dynamic interactions in the workplace [1]) are better captured through the application of graph-based centrality metrics as several studies show [1,6,11–14,23,26].

The formulation and application of graph-based theory to understand the myriad of interrelationships between dynamic entities is not new and spans agriculture, anthropology, project management, biology, economics, marketing, criminology, political, computer science, and organizational studies—just to name a few [27–29]. Table 1 shows some of the most notable developments of graph-based metrics across the recent years.

**Table 1.** Development of graph-based centrality metrics.

| Year | Event | Description |
|---|---|---|
| 1930–1953 | Formulation of graph theory by Jewish-Hungarian mathematician Dénes König. | Publishing of the König's Book—*Theorie der endlichen und unendlichen Graphen*—in USA. König's ideas started to be developed by Haary and Norman, and since then, began to be applied to study Social Networks [27–32]. |
| 1940–1950 | Development of three most important graph-based metrics: (1) In-degree, (2) out-degree, and (3) total degree. | Psychologists Leavitt, Bavelas, and Smith in 1950 developed three of the most popular centrality measures [1,4,8]. These are used to measure how many links or preferences one entity (person, group, or organization) receives or gives from or to other actors of the social network where they exist. |
| 1950–1970 | Development of Betweenness centrality. | Started to be developed in the late 1940's by Cohn and Marriott, was finalized in the late 1970's by Anthonisse (1971), Freeman (1977), and Pitts (1979) [9,33]. Betweenness Centrality calculates the shortest path between every pair of nodes in a connected graph, and it can be used to describe the amount of influence that an entity has over the flow of information in a network. It is also often used to find entities that serve as a bridge between two different blocks of a network [9,33]. |
| 1960–1975 | Development of Closeness centrality. | Closeness centrality was developed followed the works of Bavelas in 1950, Harary in 1959, by Beauchamp in 1965, and Sabidussi in 1966, and finalized by Moxley and Moxley and Rogers in 1974. It measures how close one entity is to all the other entities within a network [9,33]. |
| 1970–1980 | Development of Density. | Another popular centrality measure that are used to characterize group cohesion is the Density. Started by Bott in 1957 and finalized in 1980 by Thurman, it represents how strongly or how weakly a network is connected regarding the number of links between entities, which represents how far an entity can reach another entity through a set of intermediate links [9,33]. |
| 2000–2017 | Redevelopment of the centrality concept in graph-based theory | The latest research argues that the centrality concept needs to be revised and should not be uniquely dependent on the position of an entity in a network as Sabidussi in 1966 and Freeman in 1979 proposed [33,34]. Such redevelopment stated that some centrality metrics in isolation may be inefficient to explain dynamic behavior. They argue that the nature of work that the entities execute should also be added when analyzing dynamic behaviors or that centrality metrics should be supported by some type of other metrics from other scientific areas [33,34]. |

As it can be seen in Table 1, the redevelopment of centrality metrics suggests that existing centrality metrics in isolation do not efficiently characterize measured dynamic behavior (informal relationships). There is a need to support existing graph-based centrality metrics in order to extract as much as possible insightful information when analyzing the interactions of entities within a network. This can be done with the modification of existing graph-based centrality metrics and with the development of new metrics.

However, research and development have been conducted in recent years not only on centrality measures. A very popular field called formation theory uses predictable models based on graph-based centrality and dispersion metrics to generate random networks, which produce graphs with small-world properties that mirror how certain actual parameters function, as well as how future relationships will evolve and emerge in the future [35,36]. Such models assume that network formation is based on a probability of attachment between any two entities called a preferential attachment mechanism, which is the basis of almost 80% of all relationships [35,36].

Applying graph-based centrality metrics in project management is still in a very initial phase, however, research shows that there are some considerable benefits of its application. For example, researchers argue that centrality in project management can be a measure of prestige, importance, influence, and control [1,34,35]. Centrality can still be used as a measure of coordination and collaboration performance in project management tasks and activities' execution, for example [37]. Research also shows that centrality plays a key role in project decision-making. For example, research shows that network position strength (known as centrality) and network tie strength (known as familiarity) have a positive effect on project decision-making [38].

In this work, we address the suggestion from [33,34] by adding other scientific areas to existing centrality metrics, namely statistics, developing new centrality metrics based on graph theory, and still reinforcing the application of centrality metrics in project management to measure individual and collective performance, as suggested by research, by analyzing five most voted relational dimensions that emerge and evolve among project management stakeholders.

### 3. Materials and Methods

In this section, we will introduce and develop the five key collaborative dimensions (5-KRD), the respective proposed metrics for each one of the five dimensions, and the necessary data to be collected (PEICs—project exchange information channels) for each one of the proposed metrics. The graph-based metrics proposed in this work comprise existing, adapted, and new metrics to analyze the 5-KRDs that emerge and evolve as people work together across different phases of a project lifecycle. These five key dimensions result from a survey conducted between 2018 and 2021 with 700 international project stakeholders from eight different business sectors, where participants, among other questions, were asked to name some of the most important dimensions of collaboration in project environments. From 700 surveyed project stakeholders, 558 valid answers were obtained, wherefrom the most voted project relational dimensions form the abovementioned 5-KRDs. These are: Construction sector with 48% valid answers, Food and Beverage with 24% valid answers, IT with 21% valid answers, Cosmetics with 12% valid answers, Life Sciences with 10% valid answers, Banking with 9% valid answers, Healthcare with 7% valid answers, and Car Industry with 4% valid answers. The five KRDs are: (1) Communication, which characterizes how different entities within a given project social network, communicate, and the respective insight as to what is being communicated, (2) internal and external collaboration, which characterizes how entities within and between organizations or departments exchange information needed to carry out activities or tasks to accomplish project related tasks and activities, (3) know-how exchange and power, which characterizes how know-how is being created and shared, and the informal power that is exerted between the entities of a group, or different groups, (4) clustering (variability effect), which characterizes how relationships are developed across a bounded and finite period of time, between entities of a group, or different groups regarding group cohesion degree, and finally (5) teamwork performance, which characterizes to what extent team effectiveness is related with efficient execution of project-related tasks and activities. Table 2 illustrates a detailed description of each one of the 5-KRDs that will be analyzed in this work.

Thought communication and collaboration can be considered high-level dimensions in projects (meaning that they may contain other sub-dimensions, such as information exchange, for example), in this work, the aim is to clearly isolate the most voted dimensions (sub-dimensions) associated with communication and collaboration. This is done in order to enable a deeper insight into how relationships emerge and evolve in organizations and the impacts on how work is performed in organizations. Therefore, we divided communication and collaboration into several dimensions (presented in the survey to project stakeholders).

**Table 2.** The five critical key collaboration types (5-KRD) detailed.

| | |
|---|---|
| 1- Communication | How do project roles, such as project managers, experts, engineers, or project administrative roles, communicate and the respective consequences of such communication behavioral patterns? Topics such as reach, strong or weak feedback, and the presence of project roles in project meetings are suitable to be analyzed. |
| 2- Internal and external collaboration | How strong is the dependency level regarding project-related information between any two given project teams or groups? What level of collaboration (collaborative overload or lack of collaboration) is practiced in a project social network? |
| 3- Know-how exchange and informal power | How is project-related information shared across the different project stakeholders of a project social network? How do influential informal and usually invisible project stakeholders influence decision-making and the execution of tasks and project activities? |
| 4- Team-set variability | How does the variability of a project team-set impact project outcome? Does an unchangeable team set from the beginning until the end of a project help to achieve more project success than a continuously changing project team set? |
| 5- Teamwork performance | How is the level of project team performance measured in feedback replies regarding important project information? |

For each one of the 5-KRDs, centrality metrics are associated in order to enable the characterization of each one of the mentioned 5-KRD. For example, for the communication dimension, and because this dimension may involve many other sub-dimensions, three metrics have been developed. They are: (1) Role attendee degree, (2) internal mail cohesion degree, and (3) feedback degree. The data source for these three metrics are project meetings and project emails. For the Internal and external collaboration, the metric Information Seeking/Provide Degree was created. The data source for this metric is project emails. For the Know-how exchange and informal power dimension, the Action Key Players metric was created. The data source for this metric is project surveys (questionnaires). For the Team-set variability the Meetings Cohesion Degree metric was created, and the Teamwork performance metric was created. Detailed information regarding each one of the metrics is illustrated in the case study section. The data source for these metrics is project meetings. The necessary data to be collected in each one of the project information exchange channels (PEICs) are illustrated in Table 3.

**Table 3.** Project exchange information channels and respective required data.

| PEIC | Necessary Data for Proposed Metrics |
|---|---|
| Project Meetings (Events) | Number of conducted project meetings in each one of the phases of a project lifecycle<br>Number of participant project stakeholders in each one of the project meetings<br>Project role name and to which team the respective project role belongs |
| Project Exchanged Mails | Number of exchanged emails sent/received in each one of the phases of a project lifecycle that regards project related information. Emails are organized as follows:<br>- *Emails sent seeking help regarding project tasks and activities*<br>- *Emails sent providing help regarding project tasks and activities* |
| Project Surveys (Questionnaires) | Conduct a simple social network analysis assessment by applying pre-defined questions that uncover important project-related information. Questions can be as follows:<br>- *Question A: Whom do you turn to, to get information or help concerning project-related issues that is important to execute your project's activities and tasks?*<br>- *Question B: Whom do you ask for permission/approval or advice regarding the starting of the execution of project tasks and activities, even if these have been already previously communicated?* |

Regarding the application of the seven proposed metrics, a specific framework that guides the implementation and calculation process of each one of them throughout all the existing phases of a given project lifecycle is illustrated in Figure 1.

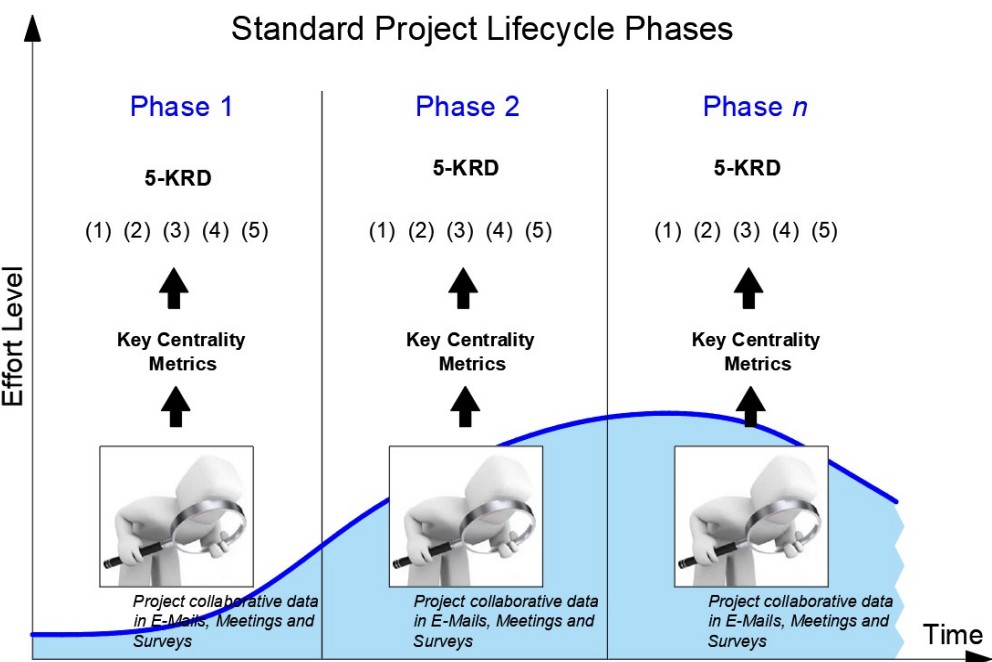

**Figure 1.** Data collection and analyzing process.

Figure 1 shows the data collection and analyzing process of the developed graph-based centrality metrics proposed in this work. There are essentially four steps. In step 1, the phases of a project lifecycle must be clearly identified. In step 2, necessary data according to Table 3 must be collected. In step 3, a set of graph-based centrality proposed metrics will be applied to the collected data. Finally, in step 4, the results of the application of the proposed metrics will be used to characterize each one of the 5-KCDs.

## 4. Development and Application of Proposed Metrics

### 4.1. Proposed 7 Graph-Based Centrality Metrics

Table 4 illustrates the seven metrics proposed in this work, the correspondent five key relational dimensions, the data source of each one of the metrics (PEIC—Project exchange information channels), and the main and auxiliary measurements, which correspond to the first and second measurement, respectively.

As an example, to characterize the internal and external collaboration relational dimension, only email project-related data regarding seeking and providing will be collected. Then, the collected data will be analyzed by the application of in- and out-degree (first measurement), and then by the application of basic statistics—the mode (second measurement). Finally, the results will be outputted and are ready for the interpretation steps, which may include the correlation of results with project outcomes and outputs.

### 4.2. Case Study

The seven proposed metrics in this work will be explained in detail, supported by a case study conducted in a life science organization in mid-Europe in 2021. In this work only, the part regarding the calculation process of the key metrics will be illustrated. A small organization (<80 workers) applied the proposed graph-based centrality metrics in an R&D project with a duration of six months, in order to uncover in a quantitatively way the myriad of relationships that emerged and evolved across the execution of the project. The project is named project P1 and concerns the development of a new endoscopy equipment part to be assembled in the final endoscopy equipment. Figure 2 illustrates the full project lifecycle as well as the participating stakeholders.

**Table 4.** The seven graph-based metrics proposed in this work.

| Sources Metrics | PEICs | | | First Measurement | Second Measurement | 5-KRD (Five Global Collaboration Types) |
|---|---|---|---|---|---|---|
| | Data from Meetings | Data from Mails | Data from Questionnaires | | | |
| Metric M1-Role attendee degree | x | | | SNA: Total In-degree | Statistics: Linear Regression | Communication |
| Metric M2-Internal mail cohesion degree | | x | | SNA: Total degree and Density | Statistics: Average | |
| Metric M3-Feedback degree | | x | | SNA: In-degree and Out-degree and Reciprocity | Statistics: Mode | |
| Metric M4-Information Seeking/Provide degree | | x | | SNA: In-degree and Out-degree | Statistics: Mode | Internal and external collaboration |
| Metric M5-Action key players | | | x | SNA: Total In-degree | Statistics: Mode | Know-how exchange and informal power |
| Metric M6-Meeting's cohesion degree | x | | | SNA: Average weighted total-degree | Statistics: Linear Regression | Team set variability |
| Metric M7-Teamwork performance | x | | | SNA: Total In-degree | Statistics: Average | Teamwork performance |

x = means that for a given PEIC (meetings, mails, or questionnaires), a given metric M (1, 2, 3, . . . , 7) is used.

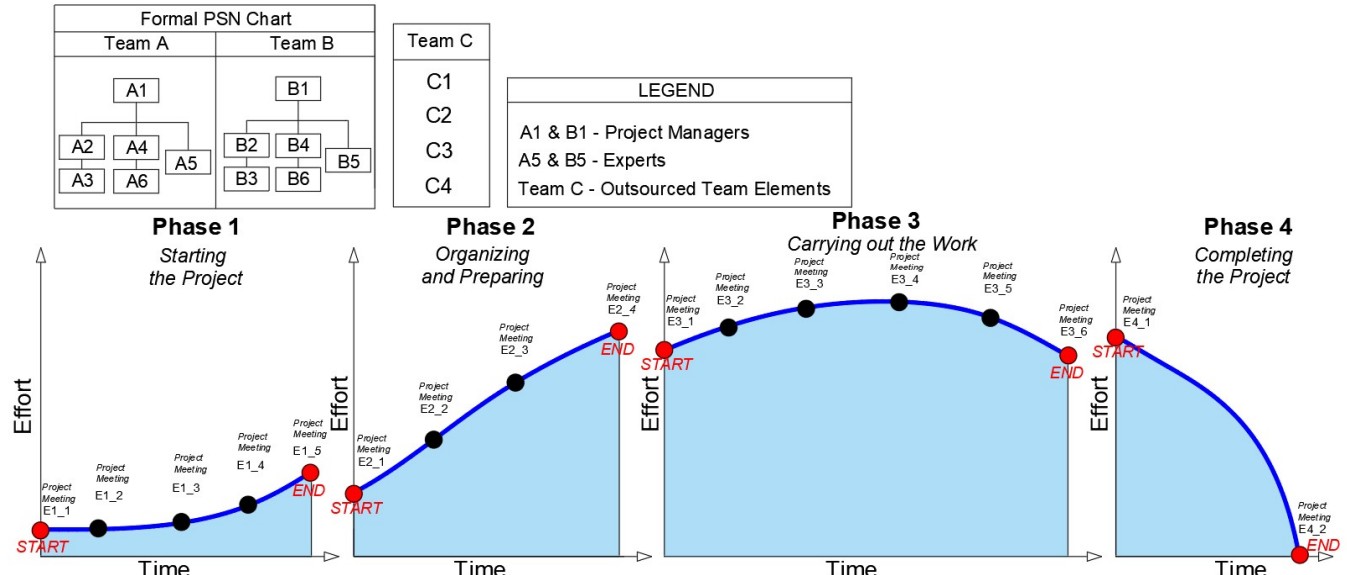

**Figure 2.** Project lifecycle phases of Project P1.

The project to be analyzed is project P1 (Figure 2) and it was delivered by two project teams (team A and team B) of the life sciences organization that collaborated across the P1's four project phases.

In this work only, a reduced part of the information captured in the whole assessment will be illustrated due to restrictions. However, the calculation process is affected by this reduction.

The Official Project Roles designated to accomplish the project P1 are displayed in the upper left corner of Figure 2 (Formal PSN Chart) and disclosed in the legend above the project lifecycle of P1.

The project roles include project managers, experts, and outsourced team members. All the nondisclosed official project roles in this case are to be considered All, according to

Table 5. There are six official project people from both teams that are planned to accomplish project P1.

**Table 5.** Independent and collective analysis of the Project Official Roles of Project P1.

| Metrics Number | Independent PSN Stakeholders | | | Global PSN Stakeholders |
| --- | --- | --- | --- | --- |
| | Project Managers | Experts | Outsourcers | All Official Defined Project Roles (Team A and Team B) |
| Metric M-1 | x | x | | |
| Metric M-2 | x | x | | x |
| Metric M-3 | | | | x |
| Metric M-4 | | | | x |
| Metric M-5 | | | x | x |
| Metric M-6 | | | | x |
| Metric M-7 | | | | x |

x = means that for a given Independent/Global PSN Stakeholder, a given metric M (1, 2, 3, ... , 7) is used.

In project P1, in phase 3, a third party—outsourced Team C—participated in some project activities. The elements (c1, c2, c3, and c4) of Team C are identified in Figure 2 (Team C). In this phase, the data collection process took place according to Table 3. In each project phase, a set of project meetings (Events E) took place, which is illustrated in Figure 2. At the end of each one of the project phases of project P1, all the project-related email data exchanged are collected according to Table 3. Across this work, only data from phase 1, phase 2, and phase 3 of project P1 will be used for the demonstration of the calculation process of the seven proposed metrics.

Four different official project roles (OPR) will be considered. They are Project Managers, Experts, Outsourcers, and All (all other administrative project roles that take place in the execution of the project P1). The OPR will be analyzed independently and collectively (Table 5). Independently means that some official project roles will be analyzed separately across the different P1's project phases. Collectively means that either all, or combination of certain official project roles, will be analyzed throughout all the different phases of a project lifecycle.

For example, for the metric Internal mail Cohesion (M-2) degree displayed in Table 5, there are two types of analysis. The first one is an independent analysis of the project managers and the experts in an isolated mode. The second is a global analysis (collectively) of all the project stakeholders (also known as project people) that participated in a project, which includes all the officially defined project roles.

4.2.1. Role Attendee Degree

Description: This metric captures the presence of two important stakeholders (Project Managers and subject matter Experts) in project meetings throughout a given project phase across P1's project lifecycle. It calculates a trend line (across a phase of a project), positive, negative, or constant, regarding the participating rate of the desired OPR. In this work, project managers and experts from both teams will be analyzed.

Method: The presence of the OPR across a project phase will be recorded and plotted in a cartesian graph (Figure 3b), where the attendance in one project meeting will be given a value of 1, and non-attendance will be given a value of 0. After that, a linear regression will be calculated (trend line), and the evolution signal will be calculated. There are four possible outcomes (Figure 4). They are: (1) Increasing (positive evolution across a given project phase), (2) Decreasing (negative evolution across a given project phase), (3) Full (constant evolution across a project phase, where the chosen OPR participated in all project meetings across a project phase) and (4) Neutral (constant evolution across a project phase, the chosen OPR did not participate in all project meetings across a project phase).

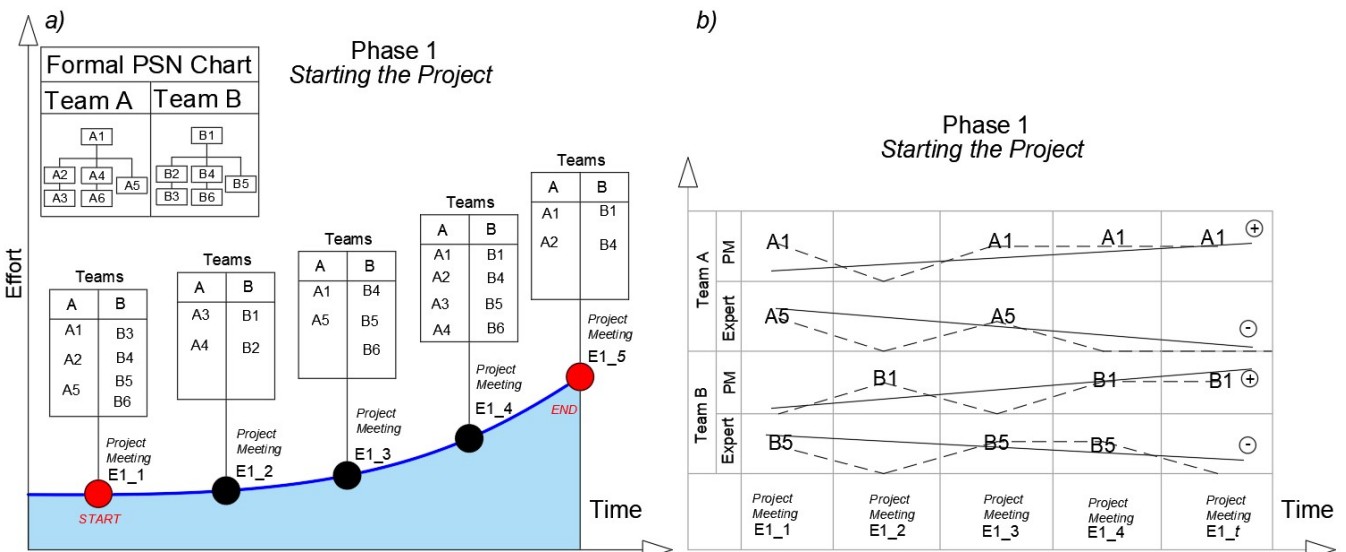

**Figure 3.** Role Attendee Degree Calculation for Project P1. (**a**) PLC of phase 1 and respective project meetings; (**b**) Evolution across Phase 1 regarding PMs and Experts from teams A and B.

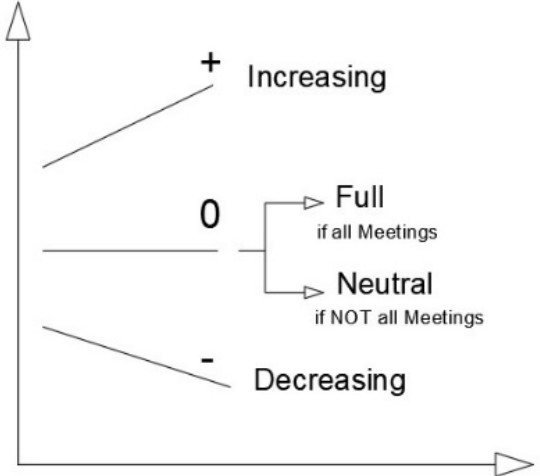

**Figure 4.** Role Attendee Degree Evolution Outputs.

    <u>Uncovers</u>: What type of evolution (increasing, decreasing, full, or neutral) based on the Role attendee degree in project meetings within a project phase, exists with higher frequency.

    Figure 3a illustrates phase 1 of project P1.

    In this phase, there were five project meetings (E1_1 up to E1_5) where project participants of two teams worked together from the beginning until the end of phase 1. From Team A, project element A1 is the project manager and element A5 is the expert. In this case, there is only one expert in each of the Teams A and B. However, multiple roles, as for the expert's case, are supported by the proposed metrics. The other elements are Engineers from different areas, such as Processing, Designing, Automation, etc. In Team B, project element B1 is the project manager and B5 is the expert. The same goes for the remaining elements of Team A. Figure 3b presents how the metric Role Attendee Degree is calculated. As an example, project element A1 participated in four out of five meetings that took place in phase 1 of project P1. A1 participated in the E1_1 (value = 1), E1_3 (value = 1), E1_4 (value = 1), and the E1_5 (value = 1) meeting. Project element A1 did not participate only in the E1_2 (value = 0) meeting. This evolution is seen in Figure 3b, as the black line indicates the real evolution regarding element A1, and the blue line is the resulting evolution of time outputted by applying linear regression across the meetings that element A1 participated

and did not participate. The resultant evolution is positive (+). This means that element A1 had higher constant participation in project meetings in phase 1, as phase 1 was nearing its end. The same applies to the other three OPRs—A5, B1, and B5.

As mentioned, there are four possible outcome types for this metric (Figure 4). They are: Increasing (positive evolution), Decreasing (negative evolution), Constant Full (participation in all project meetings), and Neutral Constant (participation in some project meetings).

In the case of project P1 (Figure 3), the project manager of Team A (A1) had a positive evolution (+), whereas the expert of Team A (A5) had a negative evolution (−) across phase 1. For Team B, the project manager B1 had a positive evolution (+), and the expert B5 had a negative evolution (−) across phase 1 of project P1.

### 4.2.2. Internal Mail Cohesion Degree

Description: In the first approach, this metric captures the percentage of project people from both Teams A and B, that were involved in all email communication that concerns project-related information across a project phase of a project lifecycle. For this metric, all mails that were sent/received directly to, forwarded to, or in CC to, will be used as input for the metric. As an example of application, (first approach) this metric aims to calculate the email communication cohesion degree, within the project Team A. In the second approach, it will analyze the Total Degree (in-degree + out-degree) of the two already named OPR (Project Managers and Experts).

Method: In the first approach, the density of the email communication network will be calculated. All emails that contain project matter information related sent and received by elements of Team A, will be collected and analyzed. A graph will be created to illustrate the mail communication, and the density metric will be calculated according to (1) (adapted from [9]).

$$d = \frac{2LM}{NM(NM-1)} \tag{1}$$

where:

*LM = total number of existing links at the email communication network*
*NM = total number of project people connected, within the email communication network*

Uncovers: To what extent does not being in all email communication network that relates project information across a given project phase that officially (according to formal chart) belongs to a project influence project outcome?

Figure 5 represents phase 1 of project P1. In the upper right corner, the graph inside the box that contains Team A and Team B represents the email communication network that contains all the exchanged project-related emails across phase 1 of project P1. For the first approach, the cross-boarded (from Team A to Team B, and within Team B) emails exchanged will not be analyzed. For example, it is visible that there is not a link between A1 and A2. This means that across phase 1 of project P1, there was no single email directly exchanged between A1 and A2 concerning project-related matter. The same happens, for example, between elements A6 and A1.

This shows, according to the email communication network, that there has been information that might have not been fully shared with all the elements of project Team A. If there was a link between all elements of Team A, this could mean that all information had been shared with all the project people of Team A. This does not necessarily mean, however, that all information has not been shared across all the elements of Team A. For example, the information that flowed in the links between A1 and A3, A1 and A5, and A1 and B1, might have been forwarded by A3, to A2, A5, and A4. However, that might, or might not have taken place, and if yes, it still might have occurred with a certain time delay. In this case, the proposed metric, due to privacy and legal constraints, does not enable further disclosure.

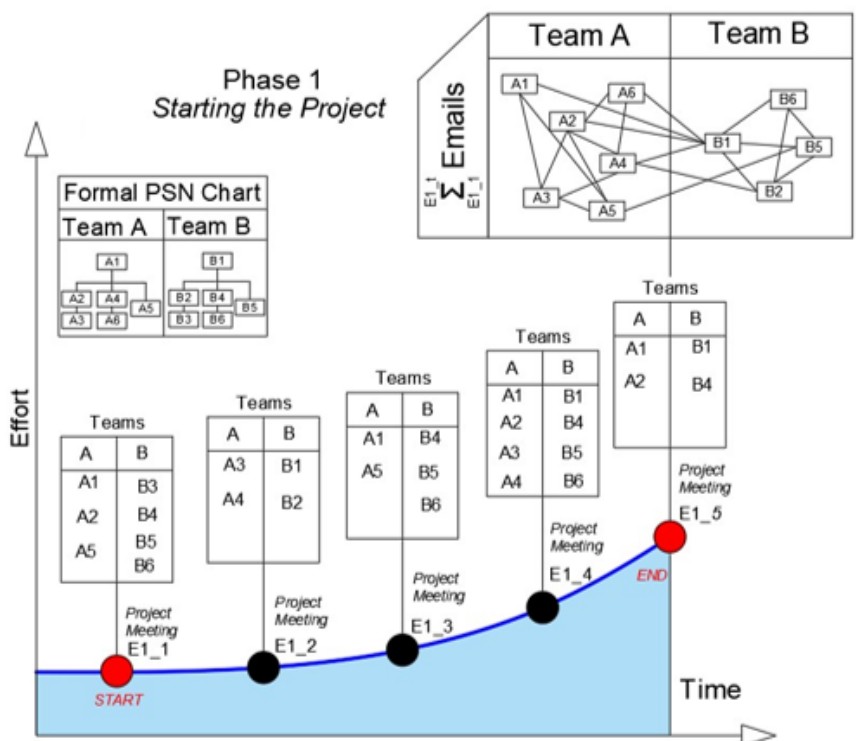

**Figure 5.** Internal mail cohesion degree calculation for Project 1.

First Approach

To measure the email communication network cohesion degree, the density (1) will be used. The maximum value for the density is when all elements of Team A have a link between them. Applying (1):

$$d = \frac{2 \times 9}{6(6-1)} = 60\%$$

These results show that through the mail communication network, 60% of all possible connections (100%) are present in phase 1.

Second Approach

In the second approach, the total degree (2) will be measured for two OPR—project managers and experts. This is illustrated in Table 6. The total degree, which represents all the links (incoming, and outgoing) that one project person has, is given by (3) (adapted from [9]).

$$T_D(NM_i) = \sum NM\_l_i \tag{2}$$

where:

$T_D$ = *total degree in the email communication network*
*NM = total number of project people connected within the email communication network*
*i = project person = 1, 2, 3, . . . , NM*
$NM\_l_i$ = *total number of existing links attached to project person i*

**Table 6.** Total Degree for Project Managers and Experts.

| | Team A | | Team B | |
|---|---|---|---|---|
| | **Project Manager (A1)** | **Expert (A5)** | **Project Manager (B1)** | **Expert (B5)** |
| Total Degree | 3 | 4 | 7 | 4 |

In Figure 5, element A1 has three links and the A5 has four links. In Team B, B1 has seven links and B5 has four links. In this case, it is clear that Team B has an advantage regarding the Project Manager total-degree, and there is an odd regarding experts of both teams. It means that that the project manager from Team B holds an advantage regarding the centrality in the email communication network, making him probably a more powerful stakeholder than the project manager from Team A, regarding the email communication network. Regarding the experts, none of them holds an advantage regarding this metric. Now, to better understand the purpose of this metric, assuming Team A as a service provider and Team B as a customer, and the email communication network as the relationship between them, it could be concluded that the project manager from the customer side holds a privileged position regarding the email communication network. The next step would be the analysis to what extent such relationship is correlated with project failure or project success.

### 4.2.3. Feedback Degree

<u>Description</u>: This metric uncovers the percentage of all project-related emails that were sent between the teams, in this case, from Team A to Team B. It does not show exactly if one particular mail has been replied to (based on its content), rather the overall number of exchanged emails. This metric aims to uncover from which side (Team A or Team B) the email communication network is more intense, which could reflect more or less control over the email network, and ultimately more or less feedback.

<u>Method</u>: For this purpose, the reciprocity metric will be used, which is simply the ratio between the emails sent from one team to another team.

<u>Uncovers</u>: To what extent does a high or a low project-related information email exchange from a given project team is correlated to a project outcome (usually success and/or failure)?

Figure 6 presents the mail communication from phase 2 of project P1 between Team A and Team B. To calculate the feedback degree, the emails sent/received from/to need to be identified. For this case, the in-degree (representing an email received) and the out-degree (representing an email sent) need to be first calculated. The in-degree, which are all the links that one project person receives, is given by (3) adapted from [9]).

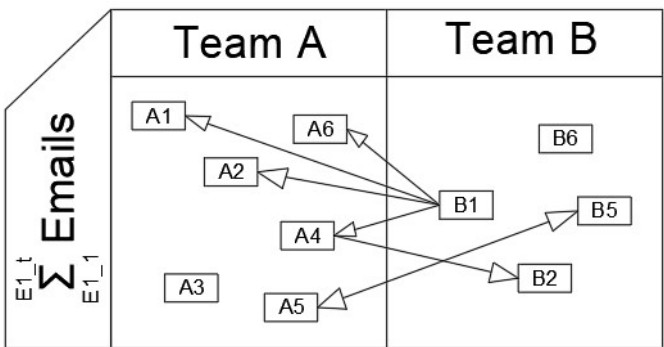

**Figure 6.** Feedback Degree Calculation for Project P1.

$$I_D(NM_i) = \sum NM_{lin} \tag{3}$$

where:

$I_D$ = *In-degree in the email communication network*
*NM = total number of project people connected, within the email communication network*
*i = project person = 1, 2, 3, . . . , NM*
*NM_$l_{in}$ = total number of existing in-links attached to project person i*

Applying (4) to elements of Team A, and Team B respectively:

$$I_D(A1) = 1$$
$$I_D(A2) = 1$$
$$I_D(A3) = 0 \qquad\qquad I_D(B1) = 0$$
$$I_D(A4) = 1 \qquad\qquad I_D(B2) = 1$$
$$I_D(A5) = 1 \qquad\qquad I_D(B5) = 1$$
$$I_D(A6) = 1 \qquad\qquad I_D(B6) = 0$$

The out-degree, which are all the emails that one project person sends to other, is calculated applying by (4) adapted from [9].

$$O_D(NM_i) = \sum NM_{lout} \qquad (4)$$

where:

$O_D$ = *Out-degree in email communication network*
*NM = total number of project people connected, within the email communication network*
*i = project person = 1, 2, 3, . . . , NM*
*NM_$l_{out}$ = total number of existing out-links attached to project person i*

Applying (5) to elements of Team A and Team B, respectively:

$$O_D(A1) = 0$$
$$O_D(A2) = 0$$
$$O_D(A3) = 0 \qquad\qquad O_D(B1) = 4$$
$$O_D(A4) = 1 \qquad\qquad O_D(B2) = 0$$
$$O_D(A5) = 1 \qquad\qquad O_D(B5) = 1$$
$$O_D(A6) = 0 \qquad\qquad O_D(B6) = 0$$

As a conclusion, Team A sent two mails to Team B, and Team B sent five mails to Team A. Total mails sent between Teams were seven. The feedback degree will be calculated by applying the reciprocity given by (5) adapted from [9].

$$RM = \frac{Sent\ Mails\ low}{Sent\ Mails\ high} \qquad (5)$$

where:

*RM = Reciprocity in email communication network*
*Sent Mails low = sum of the lowest number of emails sent by one given team*
*Sent Mails high = sum the highest number of emails sent by one given team*

Applying (6):

$$RM = \frac{2}{5} = 40\%$$

There is a 40% reciprocity in the mail communication network between Team A and B, in phase 2 of project P1. This means that only about 40% of all emails send between Team A and B during phase 2 of project P1 were replied to. In this case, three emails have not been replied to and the Team B has the highest number of emails sent and emails non-replied.

### 4.2.4. Information Seeking/Providing Degree

Description: This metric uncovers which team (Team A, or Team B) is more or less dependent on project-related information. Only the emails that contain project-related information will be analyzed. This specific type of information has to be related with seeking and providing help regarding project related activities.

Method: For this purpose, all mail communication will be assessed and filtered according to seeking and providing help regarding project-related matters. This means that the email content will have to be disclosed. Access to the email content requires the permission of the organization and the respective employees. If permission is not given then this metric is limited to analyzing the email subject, being conditioned to an individual follow-up analysis regarding the respective involved employees. Next, the ratio between mails seeking help and emails sent by those providing help will be calculated.

Uncovers: Which team is more information-dependent to execute project-related activities? To what extent is the dependency of a certain team correlated to a given project outcome (usually failure and/or success)?

The upper right corner of Figure 7 shows the email communication network between Team A and B regarding mails sent seeking for help and mails sent providing advice or help regarding necessary project information to accomplish project activities across phase 1 of project P1. Mails sent asking for help are identified with blue color. Mails providing information are identified with green color. The results regarding emails sent/received are presented in Tables 7 and 8. In this case, in-degree and out-degree metrics will be applied according to (3) and (4) to Teams A and B.

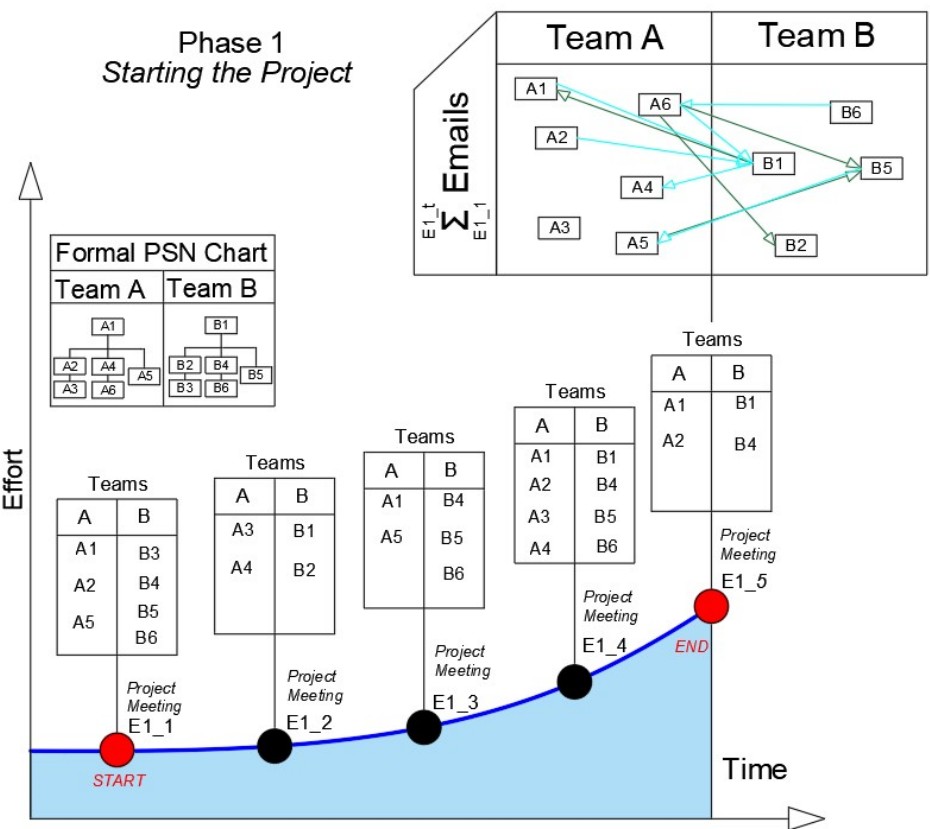

**Figure 7.** Information Seeking/Providing degree calculation for Project P1.

**Table 7.** Mails sent seeking help.

| *Blue Color* | **Team A** | **Team B** |
|---|---|---|
| Team A | - | 3 |
| Team B | 3 | - |

**Table 8.** Mails sent providing help.

| *Green Color* | **Team A** | **Team B** |
|---|---|---|
| Team A | - | 3 |
| Team B | 1 | - |

Team A sent three mails looking for help and only received one mail providing help from Team B. Team B sent three mails asking for help and Team A sent back three mails providing help. It can be concluded that both teams have the same dependency degree

(seeking), but different providing help levels or degrees. The seeking and providing reciprocity can be calculated applying (6) for both seeking and providing as follows:

$$Seeking \ R = \frac{3}{3} = 100\%$$

$$Providing \ R = \frac{1}{3} = 33\%$$

This concludes that both teams, A and B, are equally information-dependent, but Team A has a disadvantage because not all information requests were satisfied through the email communication network, which makes Team A more information-dependent than Team B. There are two possible outcomes for this metric. Either both teams are equally dependent, and in that case, the seeking and providing reciprocity degrees are 100%, or one of the teams is more or less dependent, as for example, in the case of Figure 7. Results that are equally dependent are considered neutral and, therefore, no conclusion can be outdrawn.

### 4.2.5. Action Key Players

Description: This metric is to be applied, but not only when a third party is outsourced by one of the teams, A or B, to execute project activities in a given project phase. This metric uncovers what are the key players among the elements Teams A and B that share know-how and provide guidance to the third team in order to execute project activities. In other words, it aims to identify who has the power to delegate and take decisions, and to what extent the way these decisions are taken influences project outcome.

Method: A simple graph analysis (also known as social network analysis) will be conducted addressing all elements of the third team (usually called Team C, or TC) in order to find out who the most important people are (informally) for Team TC regarding support (know-how and decision-making) so that Team TC can execute project activities which it was outsourced for. Two questions will to be asked to the Team TC in the assessment. These questions are also illustrated in Table 3. After the SNA assessment is ready, by applying SNA theory, key players will be identified. They will be essentially identified by using in-degree and out-degree, to find Central Connectors and Peripherical people within Teams A and B.

Uncovers: To what extent does the know-how transfer and decision-making power, coming either from Team A or from Team B influence a project outcome in project phases where a third team is needed/outsourced to execute project-related activities?

Figure 8 presents phase 3 of project P1. In this phase (carrying out the work), a third team (Team C) was outsourced to execute project-related activities. Social network analysis with the two strategic questions was conducted, addressing the elements of the outsourced team (c1, c2, c3, c4), and the results are displayed in the upper right corner box in Figure 8. In this case, blue lines are answers provided by Team A, and green lines are answers provided by Team B. The quantitative results for both questions are to be illustrated in Table 9 by applying (3).

According to the results in Table 9, it can be concluded that Team A has a privileged position regarding providing help, sharing know-how to execute project-related tasks. However, Team B takes control of the decision-making process when it comes to deciding what is to be done. For this metric, three different possible results for each question are possible. They are illustrated in Figure 9.

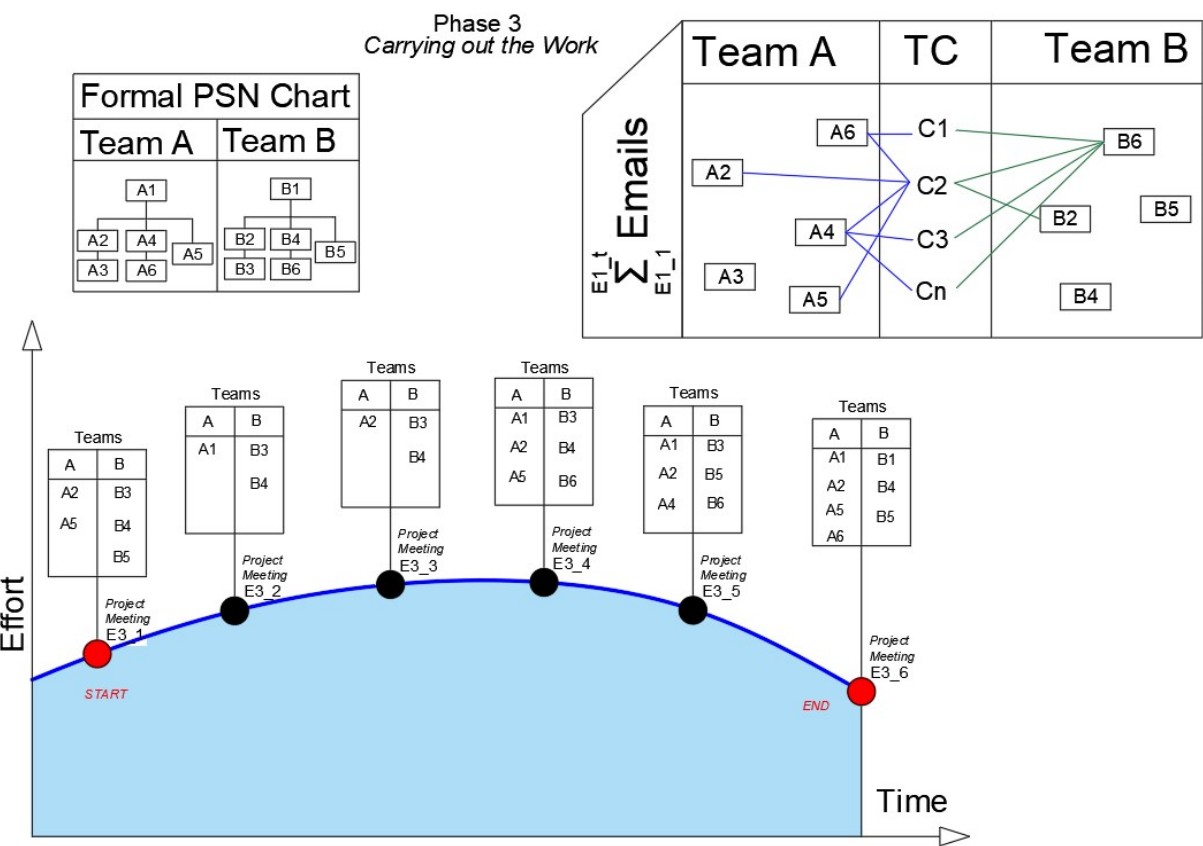

**Figure 8.** Action Key Players Calculation for Project P1.

**Table 9.** SNA Questionnaire results (In-degree) for Project P1.

| Question 1 (In-Degree) | | Question 2 (In-Degree) | |
|---|---|---|---|
| A2 | 1 | A2 | 0 |
| A3 | 0 | A3 | 0 |
| A4 | 3 | A4 | 0 |
| A5 | 1 | A5 | 0 |
| A6 | 0 | A6 | 0 |
| B2 | 1 | B2 | 0 |
| B4 | 0 | B4 | 0 |
| B5 | 0 | B5 | 0 |
| B6 | 0 | B6 | 4 |

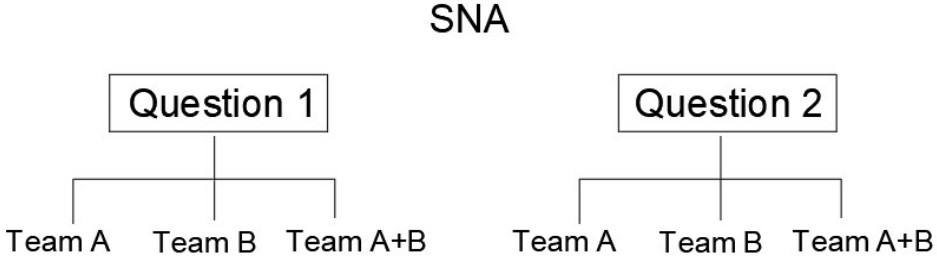

**Figure 9.** Possible outcomes for metric Action Key Players after applying a simple SNA.

#### 4.2.6. Meetings Cohesion Degree

Description: This metric was developed to quantitatively capture the complex project people variability (PSNVar—Project Social Network Variability—Figure 10) regarding the participation in F2F project meetings across a phase of a project lifecycle in a simplified and meaningful way, so that it could be translated into a single value that enables to correlate it with a project outcome. It can also be used in other project meeting environments apart from the F2F. The variability is the function of the project people that start at the very first project meeting of each project phase—they usually are at the formal chart—characterized as people that are designated to accomplish a project—phase. It includes the project people that start, leave, restart, project meetings across a project phase. This metric measures the project network social cohesion degree variation regarding meeting participation of project people across a project lifecycle based on the relationships that dynamically evolve across the project meetings of a project phase. In other words, if the same project people (project people that are officially designated to accomplish a project phase), participate in all the project meetings that occur in a given phase of a project lifecycle, a certain relationship type (project social cohesion) between them starts to emerge and keeps growing until the end of a project phase. This relationship can be translated into friendship, or simply awareness (who knows who), trust, or other, and may or may not affect a project outcome.

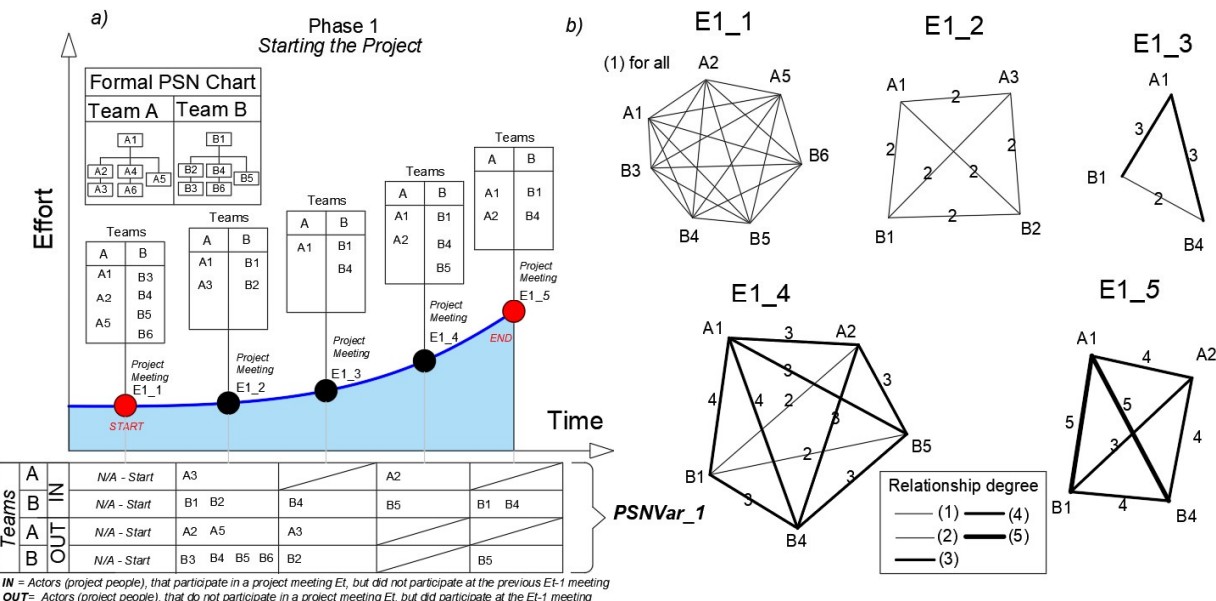

**Figure 10.** Meetings Cohesion degree calculation for project P1. (**a**) PLC of phase 1 and respective project meetings; (**b**) Different network arrangements based on participation degree in project meetings illustrated in (**a**).

Method: For this purpose, a metric was developed based on graph theory centrality metrics average in-degree. After the calculation of the variability for each project meeting, a linear regression (auxiliary measure) will be applied to find out which evolution of that line (positive, negative, or constant).

Uncovers: How does the variability (change of project people set) of project people that participate in project meetings in a project phase influence a certain project outcome? Does having an unchangeable project team set (the same elements of a project Team from the beginning until the end of a project phase), across a phase of a project lifecycle, regarding meetings participation, influence a project outcome?

Figure 10a shows project phase 1 of project P1. In this phase, there were five project meetings. In the first meeting, Team A elements (A1, A2, A5) and Team B elements (B3, B4, B5, B6) were present (this can be seen in the boxes above the dots that represent the different project meetings in Figure 10a). This means that for this metric, all these elements

met each other personally in the project P1 phase 1 context, for the very first-time regarding phase 1 of project P1. To be noted, the personal relationships or past relationships between project elements are not taken into consideration in this metric. Thus, all the project people that participated in meeting E1_1 have developed a link (relationship) with each other.

As an example, in Figure 10 b E1_1 A1 has a link to B6, which means that they were together in the first meeting of project P1 at phase 1. The first meeting is represented in a network in Figure 10b in the middle right corner as E1_1. In the first meeting, all the participants get a link from all the other participants (Figure 10b E1_1). This means that, for example, A1 has a link to all the others (A2, A5, B3, B4, B5, B6) of value 1. The value one (1) represents that they meet each other for the first time in a project meeting environment. Therefore, for A1, as for all other participants, the total in-degree at the first meeting will be 6 (all the links directed to A1). In the second meeting (Figure 10b E1_2), new elements are in and some elements that participated at meeting 1 are not there anymore. The only element that is in both meetings is A1. This means that A1 participated for the first time in a project meeting with all the other three (A3, B1, B2). Thus, A1 gets a link of value 1 from each of the others, which makes a total sum of three.

In the third meeting (Figure 10b E1_3), A1 reencounters B1 and B4. A1 and B1 were already together in the first meeting, thus now the link between them is of value two, which represents the second time that they are together in a project meeting. The same happens with A1 and B4. On the other hand, B1 and B4 meet each other for the first time in the third meeting. Therefore, the link between them is of value 1. The same principle is applied to all other project meetings (Figure 10b E1_4 and E1_5).

Figure 10a, under the project curve (blue line), shows a matrix which contains the project people variability regarding project meeting participation degree. Considering meeting E1_3 as the present meeting, elements A1, B1, and B4 participated in the respective project meetings. Element B4 is in the matrix categorized as IN. This means that element B4 did not participate in previous project meetings. Element A3, for example, on the other side is categorized as OUT. This means that he participated in the previous meeting but is not taking part in the present project meeting. The metric (6) developed to measure this variability measures exactly the number of times the same project people were together across the project meetings of a project phase. If the same people were always together in all the project meetings, the metric will output a constant value across time. If a change in the set takes place, the metric will immediately react and output a non-constant value for each project meeting.

$$V_{(Et)} = \frac{WL_{(Et)}}{TPP_{(Et)} \times Et} \qquad (6)$$

where:

*V = Variability of a PSN (project social network)*
*Et = Meeting (event) number, where Et = 1, 2, . . . , TE*
*TE = Number of project meetings (events) that occurred in a given project phase TPP = Number of project person that participated in an event Et.*
*WL = Value of all weighed connections (links), from each project stakeholder total degree in each project meeting (event) Et.*

Applying (7), to all the meetings (Events):

$$V_{E1}(Teams\ A,\ B) = \frac{6+6+6+6+6+6+6}{7 \times 1} = 6$$

$$V_{E2}(Teams\ A,\ B) = \frac{3+3+3+3}{4 \times 2} = 1.5$$

$$V_{E3}(Teams\ A,\ B) = \frac{4+3+3}{3 \times 3} = 1.11$$

$$V_{E4}(Teams\ A,\ B) = \frac{10+7+7+9+7}{5 \times 4} = 2$$

$$V_{E5}(Teams\ A,\ B) = \frac{11+8+10+9}{4 \times 5} = 1.9$$

This metric outputs three different possible results (evolutions across time). They are: (1) Non-constant evolution positive (+), (2) non-constant evolution negative (−), and (3) constant evolution (0). Non-constant positive (+): Change in the project team set (possible increasing of new project elements). Non-constant negative (−): Change in the project teams set (possible decreasing of project elements). Constant evolution (0): No change in the project teams set (constant across all project meetings). For the example illustrated in Figure 10a, the respective evolution is represented in Figure 11. It shows that a negative evolution has occurred, which means that the resulting project team (Team A and Team B) has not been the same set from the beginning of that phase until its end. Furthermore, this evolution indicates that the participant numbers have been generally decreasing across the project meetings of that project phase in project P1.

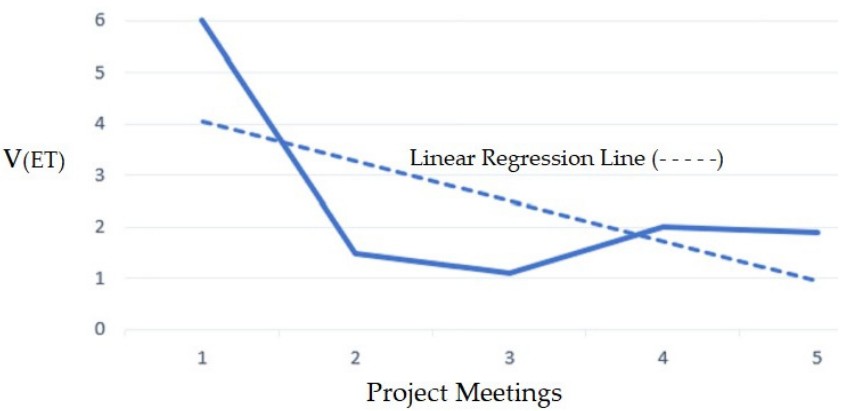

**Figure 11.** Linear regression evolution for Meetings Cohesion degree calculation for project P1.

### 4.2.7. Teamwork Performance

Description: This metric analyzes project-related information transferring speed (Average Feedback Speed when providing an answer to a project-related information question), of all Official Project Roles in all project-related email exchanges. This aspect in project management was named as a very critical one, being in fact the one that most respondents connected with efficiency within a given PSN, therefore, it is named teamwork performance.

Method: Average value of all feedback times of replied to emails within a project phase. This metric outputs an average hours-value that spans from 100% (which stands for an instantaneous reply that was made in less than 1 h) to 0% (stands for no feedback found across the duration of a project phase).

Uncovers: Feedback speed when proving an answer to a project-related question.

Figure 12a illustrates the email communication network between team A and team B only in phase 1 of project P1, and the respective duration in hours (Figure 12b). Only emails sent between different teams are illustrated in the links between the different project people in Figure 12a. The number of emails sent and received regarding a certain project-related subject is marked in yellow. For example, in phase 1 of project P1, project people A1 exchanged two emails with project people B1. This means that A1 asked B1 two times for project related information, and B1 replied to A1 two times. For this case, if an email has not been replied to within 480 h, it gets a value of 0, which means that no reply was made within the email network communication. If an email has been replied to in less than 1 h, it has a value of 100% (1). For each email interaction presented in Figure 12a, the resulting feedback time is illustrated in Table 10. For example, the two mails that A1 sent to B1 (this can be seen in Figure 12a in the yellow box line from A1 to B1) asking for project-related information (Figure 12a), both had a feedback time of 1 (100%). This means, that B1 replied to both emails from A1 in less than 1-h time period. In another example, the feedback first email time (0.3) between A2 and B1 is of 0.3 h, which means that it took 336 h to reply, on average. Two types of answers are defined. They are: (1) Instantaneous answer (1, or 100%):

Email has been replied to in less than 1-h period time, and (2) Infinite answer (0): Email has not been replied to within a project phase period time. In Figure 12b is illustrated the full duration in hours of phase 1 of P1.

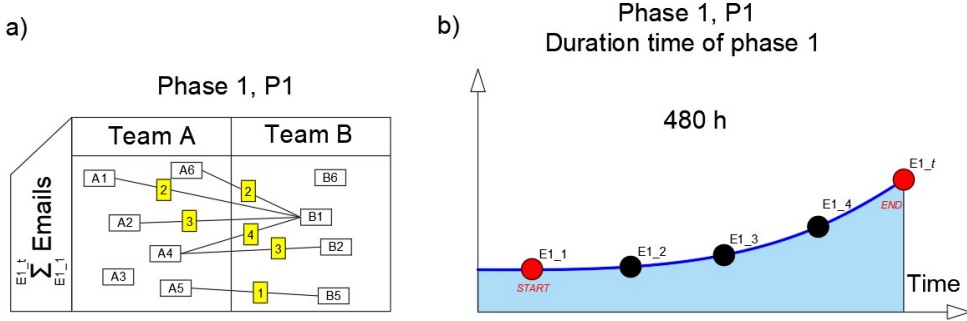

**Figure 12.** Teamwork performance Calculation for project P1. (**a**) Exchanged emails between elements of team's A and B in phase 1; (**b**) Duration time of phase 1 in hours.

**Table 10.** Interaction time Feedback.

|  | **B1** |  |  |  |  |  | **B2** | **B5** |
|---|---|---|---|---|---|---|---|---|
| A1 | 1 | 1 | ------ | ------ | ------ | ------ | ------ | ------ |
| A2 | 0.3 | 0.4 | 0.8 | ------ | ------ | ------ | ------ | ------ |
| A4 | 0.1 | 1 | 0.3 | 1 | 0.3 | 1 | 0.8 | ------ |
| A5 | ------ | ------ | ------ | ------ | ------ | ------ | ------ | 0.1 |
| A6 | 0.2 | 0.8 | ------ | ------ | ------ | ------ | ------ | ------ |

To calculate the transfer average speed, the following formula is to be applied (7):

$$Ts = \frac{\sum t_{a-b}}{TCMs} \tag{7}$$

where:

$\sum t_{a-b}$ = *Sum of all times from all replied to emails within a project phase.*
*TCMs = Total of emails sent and replied within the email network communication*

Applying (9) to Figure 12a:

$$Ts = \frac{1+1+0.3+0.4+0.8+0.1+1+0.3+1+0.3+1+0.8+0.1+0.2+0.8}{15} = 0.61$$

This means that, on average, all the emails that have been replied to within the email communication network between the two different teams had an average of 187 h, respecting phase 1 of P1.

## 5. Discussion

Across the previous section, the calculation process of the seven proposed metrics was illustrated in detail with the support of a case study. As it can be seen, the metrics efficiently capture in a quantitative way dynamic behaviors that emerge and evolve as a project moves to the end. The calculation process is simple and intuitive, nevertheless, it provides unique information regarding the hidden behavioral patterns that exist in every organization out there. The examples given across the case study are of low complexity in terms of calculation, however, they serve the purpose of illustrating the calculation process in a step-by-step approach. Nevertheless, due to GDPR restrictions, some metrics cannot go further in uncovering critical information to characterize a given key collaborative dimension. For example, in the internal mail cohesion degree metric, it is not possible to clearly analyze the content of email exchanged information to access what information

did flow between any two entities or groups, for example. This fact represents a certain drawback regarding the output of this metric. Nevertheless, if permission is given to access the content of exchanged information, then this issue is no longer a problem.

Regarding the role of attendee degree, the action key players, and the teamwork performance metrics, this issue is not a problem because there is no need to access restricted information. For example, the role attendee degree does not capture sensitive information that flows across elements of a given project social network, rather checks only the presence of key project stakeholders in project meetings. In the action of key players, this issue is also not a problem. However, another problem may arise, which has to do with the veracity of data provided in the questionnaire addressed to a third-party team. More concretely, the network analysis must consider the existence of bias in the answers of the participants, which may lead to misleading results and conclusions.

The application of the proposed metrics in projects that were successfully and unsuccessfully delivered may shed light on which critical success and failure factors are responsible for leading projects to a certain outcome. This can be uncovered if organizations have the necessary data and a substantial number of similar projects (the same project phases, the same industry, etc.) to be analyzed.

In a managerial dimension, the application of these metrics represents a new approach to tackling collaborative issues that may emerge and evolve across the different phases of a project lifecycle. In fact, several studies show that if the mix of informal networks is not properly uncovered and managed, most likely it will evolve toward one of the two collaborative extremes—(1) collaborative overload, which is characterized by a disproportional collaborative state of some project stakeholders related to others of a given project social network and may lead to the emergence of information bottlenecks and information exchange delays, just to name a few, and (2) lack or nonexistence of collaboration, which is characterized by the lack of collaborative initiatives within a project social network, which ultimately may lead to the emergence of organizational silos, for example [29]. Both information bottlenecks and organizational silos are known for having drastic negative impacts on organizations.

Furthermore, the proposed metrics use a very straight-forward and simple mathematical formulation rather than a complex system of algorithms, which benefits organizations regarding the cost-benefit of the application of such metrics.

Still, organizations can use the results outputted by the proposed graph-based centrality metrics in this work to correlate project stakeholders' behavioral patterns with project outputs and outcomes. By doing so, organizations learn in a straightforward and insightful way (lessons learned), which behavioral patterns must be replicated in future projects and which of them must be eliminated or avoided in order to increase project success outcome.

Finally, organizations benefit from the application of the proposed metrics because proposed metrics generate unique and actionable know-how in the collaborative project dimensions, which very likely will give organizations a sustainable competitive advantage when compared with other organizations that do not have such insights enabled by the application of the proposed metrics. As a direct consequence, this will directly contribute to the three typical pillars of sustainability (society, profit, and environment) because working in a more efficient way enables the timely identification of project behavior patterns that very likely may lead to failure, for example, an organization can take actionable measures to avoid the heading to a failure outcome and thus saving resources (time, energy, and people) that would be needed in order to rework or redo project tasks and activities or rescope a project.

In the academic dimension, the proposed metrics in this work will enable us to better understand the people aspect implications in a project's outputs and outcomes, which may lead to the development of novel project behavioral theories and approaches in order to better and wiser manage projects. The development of new graph-based centrality metrics may also trigger research organizations to invest more in the development of new algorithms that are able to capture project behavioral patterns in a 360° approach.

## 6. Conclusions

In this work, seven graph-based centrality metrics are proposed to analyze in a quantitative way five key collaborative project dimensions that emerge and evolve across a project lifecycle. The five key collaborative dimensions result from a survey conducted between 2018 and 2021 with 700 international project stakeholders from eight different business sectors. The conducted research in this work addresses one of the three aspects (the people aspect) that project management academicals and practitioners argue as critical to better understand how projects can be successfully delivered. More concretely, the people aspect relates to the impacts of project stakeholders' collaboration on project outputs and outcomes. The research analyzes the emergence and evolution of the mix of formal and informal networks within an organizational project social network, and how these drive behavioral patterns in all the different phases of a project lifecycle. Across the case study section, the calculation process of the seven proposed centrality metrics is illustrated in a step-by-step approach. The calculation process is simple to execute, and the results uncover unique and insightful behavioral patterns that help to characterize the five different key collaborative project dimensions.

Regarding future research, we suggest three critical areas. First, regarding the data collecting methods, further research should be conducted in order to develop collecting methods that minimize or eliminate bias as participants are responding to project surveys. In this respect, research should be undertaken in order to develop strategic questions and counter-questions that allow the researcher to identify any type of potential bias in the provided answers. Second, research should be conducted not only in the development of new and adapted graph-based centrality metrics to better mirror the myriad of informal relationships that emerge and evolve as people work together to deliver project objectives but also graph-based dispersion metrics. Finally, alternative data-collecting methods that capture relational data from other sources that are, until today, restrained due to the GDPR (General Data Protection Regulation) regulations, such as phone calls and corridor meetings, for example, should be developed.

**Author Contributions:** Author M.N. carried out the investigation methodology, writing—original draft preparation, conceptualization, the formal analysis, collected resources, and application. Other remaining authors contributed with the review, and validation. All authors have read and agreed to the published version of the manuscript.

**Funding:** This research received no external funding.

**Institutional Review Board Statement:** Not applicable.

**Informed Consent Statement:** Not applicable.

**Data Availability Statement:** Not applicable.

**Conflicts of Interest:** The authors declare no conflict of interest.

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
