# Peer review of "Creating Actionable and Insightful Knowledge Applying Graph-Centrality Metrics to Measure Project Collaborative Performance"

_sustainability, doi:10.3390/su14084592_

Round 1

Reviewer 1 Report

In Chapter 3.2.4 (Information Seeking / Providing degree), the authors evaluate the mutual communication between team A and B. I quote: “Team A sent three mails looking for help and only received one mail providing help from Team B. Team B sent three mails asking for help and Team A sent back three mails providing help. It can be concluded that in this case that both teams have the same dependency degree (seeking), but different providing help levels or degrees. ”It is just a quantitative assessment of mutual ties. It is not known whether the authors did not analyze the content of these emails and the nature of the aid requested. Therefore, it is necessary to perceive the results (evaluation) in a certain perspective or to state this clearly.
The article is strongly supported by several illustrative pictures, I rate it very positively.
There are formal errors in the article:
Correct the missing dots at the end of the sentence in multiple places.
Correct the designation 201-2020, paragraph 2. Literature Review, line 10. The error is repeated in Chapter 2. Conclusions, Implications and Further Research, line 4.

Author Response

Rev 1

Thank you very much for all your comments. We believe that in fact helped greatly to increase the overall quality of the manuscript. All your comments were taken into consideration in the revised version of the manuscript. All changes area marked in red in the new manuscript version.

In Chapter 3.2.4 (Information Seeking / Providing degree), the authors evaluate the mutual communication between team A and B. I quote: “Team A sent three mails looking for help and only received one mail providing help from Team B. Team B sent three mails asking for help and Team A sent back three mails providing help. It can be concluded that in this case that both teams have the same dependency degree (seeking), but different providing help levels or degrees. ”It is just a quantitative assessment of mutual ties. It is not known whether the authors did not analyze the content of these emails and the nature of the aid requested. Therefore, it is necessary to perceive the results (evaluation) in a certain perspective or to state this clearly.

Thank you very much for this point. This point is clarified in the method of the respective metric in 3.2.4. However, we clarified a bit more on this topic in the respective section.

The article is strongly supported by several illustrative pictures, I rate it very positively.
There are formal errors in the article:
Correct the missing dots at the end of the sentence in multiple places.

Thank you very much for this point. All sentences have been checked against this issue and updated.

Correct the designation 201-2020, paragraph 2.

Thank you very much for this point. Correction done.

Literature Review, line 10. The error is repeated in Chapter 2.

Thank you very much for this point. Correction done.

Conclusions, Implications and Further Research, line 4.

Thank you very much for this point. Correction done.

Reviewer 2 Report

The paper addresses issues related to measuring organizational performance, in this case the interaction between employees through the application of graph-based theory.

The set of graph-based centrality metrics to quantitatively measure relational data in a project environment is presented in detail. The application of the proposed method is also illustrated in a case study.

There are a number of typos:

In quoting the bibliography instead of [9, 10, 11, 12], it is recommended [9-12].

Line 70- editing error - tehri organization – their organization

Line 106- editing error - state pf the art - state of the art

Line 235 – Tabel 1 should be replaced with table 3

More editing effort is necessary in Tabel 3 and paper in general.

Line 246 - Caption in figure 2 is not proper. Also Figure 3 – line 311.

Line 273 - Internal mail Cohesion degree is refered as displayed in Table 4 but there is no information about it.

Line 278 - Tabel 4 – There should be data in the table field that exemplifies the usefulness of the presentation.

Line 334 – Formula 1 is worded incorrectly.

For a good credibility of the proposed method it would be useful to report a validation in practice.

It would be useful to have a discussion chapter in which to discuss the results obtained in relation to other results reported in the literature.

The references does not comply with the requirements of uniform writing.

Author Response

Rev 2

Thank you very much for all your comments. We believe that in fact helped greatly to increase the overall quality of the manuscript. All your comments were taken into consideration in the revised version of the manuscript. All changes area marked in red in the new manuscript version.

The paper addresses issues related to measuring organizational performance, in this case the interaction between employees through the application of graph-based theory. The set of graph-based centrality metrics to quantitatively measure relational data in a project environment is presented in detail. The application of the proposed method is also illustrated in a case study.

There are a number of typos:

In quoting the bibliography instead of [9, 10, 11, 12], it is recommended [9-12].

Thank you very much for this point. Updates done across the whole document

Line 70- editing error - tehri organization – their organization

Thank you very much for this point. Correction done

Line 106- editing error - state pf the art - state of the art

Thank you very much for this point. Correction done

Line 235 – Tabel 1 should be replaced with table 3

Thank you very much for this point. Correction done

More editing effort is necessary in Tabel 3 and paper in general.^

Thank you very much for this point. There was an importing error as we uploaded the document for the first time. We communicated this to the journal. In this new version we revised all the formatting of tables, pictures and formulas.

Line 246 - Caption in figure 2 is not proper. Also Figure 3 – line 311.

Thank you very much for this point. Correction made.

Line 273 - Internal mail Cohesion degree is refered as displayed in Table 4 but there is no information about it.

Thank you very much for this point. The metric internal cohesion degree is metric M2. This information has been added into line 273.

Line 278 - Tabel 4 – There should be data in the table field that exemplifies the usefulness of the presentation.

Thank you very much for this point. There was an importing error as we uploaded the document for the first time. We communicated this to the journal. In this new version we revised all the formatting of tables, pictures and formulas. The data in Table 4 are check marks that indicate where the metrics will be used. For example, which metrics will be used to analyze independent and global PNS stakeholders.

Line 334 – Formula 1 is worded incorrectly.

Thank you very much for this point. There was an importing error as we uploaded the document for the first time. We communicated this to the journal. In this new version we revised all the formatting of tables, pictures and formulas.

For a good credibility of the proposed method it would be useful to report a validation in practice.

Thank you very much for this point. We fully agree with this point. Other reviewers also tackled this issue and suggested to divide this section in two where in the first section we explain the proposed metrics, and in section two we present a case study as a validation method for the proposed metrics.

It would be useful to have a discussion chapter in which to discuss the results obtained in relation to other results reported in the literature.

Thank you very much for this point. We fully agree with this point and therefore we added a section - Discussion where we analyze the implications of the research to theory and practice.

The references does not comply with the requirements of uniform writing

Thank you very much for this point. We revised all the references of the manuscript.

Reviewer 3 Report

The manuscript devised a set of graph-based centrality metrics for quantitative assessment and improvement of organisational project collaborative performance. The proposed approach is further applied and demonstrated via a case study.

Although this is an interesting piece of work, the following major points are required to be addressed, prior to being considered for publication: 

  • The abstract is too lenghty, and is not in line with the journal's requirements of max 200 words. Please revise, maintaining the key information on background, purpose, methodology, results and contributions.
  • Please adopt paragraphs in the Introduction section with a view to improve impact and readability.
  • The research gap being addressed is advised to be articulated more clearly. More specific and critical analyses of the extant literature will help fortifying the significance and contribution aspects of this research manuscript.
  • Inclusion of a table in the Literature Review section, would improve the impact of this section, outlining and categorising the existing works reviewed, and evidencing further the research gap that the authors are addressing. A critical analysis column in this table can further highlight the significance of this study against the related works, and provide theoretical anchoring to the manuscript.
  • Is the content of Table 4 missing/incomplete?
  • The sections of the manuscript are recommended to be introduced in the final part of the Introduction section.
  • The methods and design adopted is advised to be adequately described. The authors are recommended to step back, and prior to introduction of the metrics, outline and justify at a higher level, the steps they have taken to complete their research (Literature Review, Model Construction and Metrics, Case Study etc.). A materials and methods section after section 2 is advised to be implemented, with a view to clarify these aspects.
  • The section 4.2 is also missing. It is recommended that the authors implement a Discussion section, discussing the implications of their research to theory and practice, reviewing their findings against extant literature and providing limitations of their research. Then, it is advised that a concise, single-sectioned, Conclusions and future directions section is implemented post the Discussion.
  • There are many grammatical errors present. A thorough proofreading towards fixing various grammatical issues and typos throughout are highly recommended.

Author Response

Rev 3

Thank you very much for all your comments. We believe that in fact helped greatly to increase the overall quality of the manuscript. All your comments were taken into consideration in the revised version of the manuscript. All changes area marked in red in the new manuscript version.

The manuscript devised a set of graph-based centrality metrics for quantitative assessment and improvement of organisational project collaborative performance. The proposed approach is further applied and demonstrated via a case study. Although this is an interesting piece of work, the following major points are required to be addressed, prior to being considered for publication: 

  • The abstract is too lenghty, and is not in line with the journal's requirements of max 200 words. Please revise, maintaining the key information on background, purpose, methodology, results and contributions.

Thank you very much for this point. The abstract has been rewritten and reduced.

  • Please adopt paragraphs in the Introduction section with a view to improve impact and readability.

Thank you very much for this point. We adopted paragraphs in the introduction and across the all document to improve impact readability

  • The research gap being addressed is advised to be articulated more clearly. More specific and critical analyses of the extant literature will help fortifying the significance and contribution aspects of this research manuscript. Inclusion of a table in the Literature Review section, would improve the impact of this section, outlining and categorising the existing works reviewed, and evidencing further the research gap that the authors are addressing. A critical analysis column in this table can further highlight the significance of this study against the related works, and provide theoretical anchoring to the manuscript.

Thank you very much for this point. We rewrote the literature review section. We fully agree with this point and therefore we introduced a table illustrating existing related works and evidencing a research gap we want to address.

  • Is the content of Table 4 missing/incomplete?

Thank you very much for this point. There was an importing error as we uploaded the document for the first time. We communicated this to the journal. In this new version we revised all the formatting of tables, pictures and formulas. The data in Table 4 are check marks that indicate where the metrics will be used. For example, which metrics will be used to analyze independent and global PNS stakeholders.

  • The sections of the manuscript are recommended to be introduced in the final part of the Introduction section.

Thank you very much for this point. We added this information in the final part of the introduction.

  • The methods and design adopted is advised to be adequately described. The authors are recommended to step back, and prior to introduction of the metrics, outline and justify at a higher level, the steps they have taken to complete their research (Literature Review, Model Construction and Metrics, Case Study etc.). A materials and methods section after section 2 is advised to be implemented, with a view to clarify these aspects.

Thank you very much for this point. We fully agree with this point and therefore we added a section named materials and methods where we clarify in a high level perspective the steps taken to conduct the research.

  • The section 4.2 is also missing. It is recommended that the authors implement a Discussion section, discussing the implications of their research to theory and practice, reviewing their findings against extant literature and providing limitations of their research. Then, it is advised that a concise, single-sectioned, Conclusions and future directions section is implemented post the Discussion.

Thank you very much for these points. Section 4.2 was mistakenly named as 4.3. We corrected this issue. Still, we fully agree with this point and therefore we added a section - Discussion where we analyze the implications of the research to theory and practice.

  • There are many grammatical errors present. A thorough proofreading towards fixing various grammatical issues and typos throughout are highly recommended.

Thank you very much for this point. We conducted a proofreading across the whole document and corrected identified grammatical errors.

Reviewer 4 Report

This paper proposed some graph-centrality metrics to measure project collaboration, some comments may be helpful to improve its quality:

(1) The literature review needs to be improved. It is better to separate it into several subsections. It looks like there is previous research that has used graph-centrality metrics for dynamic interaction in an organization. What is the difference between this study and those previous ones? Why need to propose new metrics?

(2) this paper mentions five key relational dimensions (KRD): communication,  internal and external collaboration, know-how exchange & informal power, Team-set variability, Teamwork performance. Why are they? Especially, some of them are overlapped. For example, communication should contain know-how exchange & informal power. And internal and external collaboration is a wider dimension that should include communication. 

(3) If this study is a case study, this should be emphasized in the abstract and introduction section. If not, it is better to separate section 3 into two sections. one section is used to explain the proposed metrics (please provide more justifications for these metrics). The other section is a case study to validate the proposed metrics.

(4) The formatting is out of order, which makes reading hard. 

Author Response

Rev 4

Thank you very much for all your comments. We believe that in fact helped greatly to increase the overall quality of the manuscript. All your comments were taken into consideration in the revised version of the manuscript.

This paper proposed some graph-centrality metrics to measure project collaboration, some comments may be helpful to improve its quality:

  • The literature review needs to be improved. It is better to separate it into several subsections. It looks like there is previous research that has used graph-centrality metrics for dynamic interaction in an organization. What is the difference between this study and those previous ones? Why need to propose new metrics?

Thank you very much for this point. This point has already been tackled by other reviewers. We will use the suggestion of a review that suggested the creation of a table illustrating the major previous research where graphs were used in organizations to measure performance and highlight the differences or novelty of this study.

  • this paper mentions five key relational dimensions (KRD): communication, internal and external collaboration, know-how exchange & informal power, Team-set variability, Teamwork performance. Why are they? Especially, some of them are overlapped. For example, communication should contain know-how exchange & informal power. And internal and external collaboration is a wider dimension that should include communication.

Thank you very much for this point. These metrics are the result of a project survey to 700 international project stakeholders between 2018-2022. These metrics are the most voted metrics regarding the performing of joint work in organizations which includes all dimensions of collaboration (coordination, cooperation and networking). In this work we clearly want to isolate the dimensions of communication and collaboration in order to access deeper information regarding how work is performed in organizations. Therefore, we delayered communication and collaboration in to several dimensions (presented in the survey to project stakeholders). We added this information in the manuscript.

  • If this study is a case study, this should be emphasized in the abstract and introduction section. If not, it is better to separate section 3 into two sections. one section is used to explain the proposed metrics (please provide more justifications for these metrics). The other section is a case study to validate the proposed metrics.

Thank you very much for this point. We fully agree with this point. This paper is not about a case study exclusively. Therefore, we take the suggestion to divide section 3 (now section 4) in two where in the first section we explain the proposed metrics, and in section two we present a case study as a validation method for the proposed metrics.

  • The formatting is out of order, which makes reading hard. 

Thank you very much for this point. There was an importing error as we uploaded the document for the first time. We communicated this to the journal. In this new version we revised all the formatting of tables, pictures, formulas and sections.

Round 2

Reviewer 2 Report

Observations have been treated accordingly and the paper is improved. However, attention must be paid to aspects of writing, for example in the bibliography that still need to be standardized, for example the year of publication is written twice: Freeman, L. (1979). Centrality in social networks conceptual clarification. Shock. Netw. 1979, 1, 215–239

Author Response

Thank you very much once again for your points. We reviewed all the references of the manuscript and standardized the formatting according to the Journals requirements. Changes / updates are marked in red in the new version of the document.

We also conducted a new proofreading session to identify and correct all minor English spell issues. 

Reviewer 3 Report

The authors have addressed the reviewer comments in full, significantly improving the quality and impact of their research paper and thus, the manuscript is now acceptable in its present form.

Author Response

Thank you very much once again for your comment. We still reviewed all the references of the manuscript and standardized the formatting according to the Journals requirements. We also conducted a new proofreading session to identify and correct all minor English spell issues.

Reviewer 4 Report

I'm satisfied with the revision.

Author Response

(The authors gave the same response as above.)
